# Reliable Internet of Things: Challenges and Future Trends

**Mohammad Zubair Khan** [1,*][iD], **Omar H. Alhazmi** [1][iD], **Muhammad Awais Javed** [2][iD], **Hamza Ghandorh** [1] **and Khalid S. Aloufi** [3]

1 Department of Computer Science, Taibah University, Medina 42353, Saudi Arabia; ohhazmi@taibahu.edu.sa (O.H.A.); hghandorh@taibahu.edu.sa (H.G.)
2 Department of Electrical and Computer Engineering, COMSATS University Islamabad, Islamabad 45550, Pakistan; awais.javed@comsats.edu.pk
3 Department of Computer Engineering, College of Computer Science and Engineering, Taibah University, Medina 42353, Saudi Arabia; koufi@taibahu.edu.sa
* Correspondence: mkhanb@taibahu.edu.sa or zubair.762001@gmail.com

**Abstract:** The Internet of Things (IoT) is a vital component of many future industries. By intelligent integration of sensors, wireless communications, computing techniques, and data analytics, IoT can increase productivity and efficiency of industries. Reliability of data transmission is key to realize several applications offered by IoT. In this paper, we present an overview of future IoT applications, and their major communication requirements. We provide a brief survey of recent work in four major areas of reliable IoT including resource allocation, latency management, security, and reliability metrics. Finally, we highlight some of the important challenges for reliable IoT related to machine learning techniques, 6G communications and blockchain based security that need further investigation and discuss related future directions.

**Keywords:** IoT; resource allocation; latency; security; metrics





## 1. Introduction

The Internet of Things (IoT) is one of the important technologies of this era that can add automation and smartness in several sectors such as transportation, health care, industries, agriculture, energy, and infotainment. It works by deploying sensors on different devices used in these sectors, hence allowing the measurement of important real-time data. These data are transmitted to the remote servers where it can be analyzed, and intelligent actions can be taken based on it [1–15].

IoT enables many important applications, including intelligent traffic management, safety-aware autonomous driving, saving electricity usage using smart grids, remote patient monitoring, machine health monitoring, smart industrial automation, and smart home security solutions [4,5]. In the era of Industry 4.0 and 6G communications, IoT applications will revolutionize how different industries operate.

There are several use cases of IoT in these industries. It can facilitate smart usage of electricity and communication flow between devices and the grid. Related to smart transportation, IoT can provide safety to drivers and passengers. Similarly, health care has several potential benefits offered by IoT regarding patient health monitoring and early diagnosis. IoT can also ensure health monitoring of machines used in various industries, thus improving the lifetime and working of the equipment.

The three major components of IoT will be sensing, communications, and data analytics. In the sensing part, various sensors such as temperature, current, humidity, heart rate, etc., will be deployed to get regular data measurements. In the communications part, technologies such as 6G will disseminate data from sensors to the cloud servers. Finally, data analytic algorithms will be extensively used to improve the working of IoT applications.

The successful working of IoT applications relies on reliable data transmission between sensors and servers. Reliability refers to robust communication with a high packet delivery

ratio, low latency, and defense against network attacks. Each IoT application may have different Quality of Service (QoS) requirements. To realize a reliable and robust IoT network, meeting the QoS requirements is needed [16–25].

Efficient data transmission is a key challenge to ensure reliable IoT applications. This means that data are transmitted at a high data rate such that latency is within the QoS requirement. This is possible when resources such as spectrum utilization, medium access, transmit power, computation task offloading to fog nodes, etc., is optimized. Moreover, privacy and secrecy of communication, along with maintaining data integrity, are required.

In this paper, we provide an overview of research related to reliable data transmission in IoT. We first present a discussion on massive IoT and its application requirements. This is followed by four major use cases of the IoT and categorize the reliability related IoT work into four major units, i.e., resource allocation, latency management, security and reliability metrics. We discuss the recent literature review for each category above and highlight the key idea and achieved results. Finally, we present challenges that still need to be addressed to realize a reliable IoT network.

## 2. Massive IoT and Application Requirements

As the communication data rates are increasing and connectivity among devices is enhanced, many IoT applications will be implemented, thus leading to massive IoT networks. The scale of massive IoT will be billions of machines, cars, sensors all connected with the Internet. We list some of the requirements for massive IoT applications in Table 1 [7–9]. Massive IoT will support many novel applications such as autonomous driving, augmented reality-based gaming, predictive maintenance of machines, automated surgery systems, and smart grids. As far as the communication requirements are concerned, massive IoT will require ultra-reliability of the order of 99.99999%. This is especially needed for critical applications where human safety is at stake, such as safe driving and health-related operations. Moreover, these massive IoT networks can afford a latency of less than 1ms. This is needed to make sure that the data are delivered within time so that applications can make correct decisions.

**Table 1.** Massive IoT application requirements.

| Applications | Autonomous Driving<br>Augmented and Virtual Reality Based Gaming<br>Predictive Maintenance of Machines<br>Automated Surgery Systems |
|:---:|:---|
| Reliability (Packet success rate) | ≥99.99999% |
| Latency | ≤1 ms |
| Data Rate | Upto 1 Gbps |
| Wireless technology | Bluetooth<br>Zigbee<br>IEEE 802.11<br>6G<br>Cellular V2X |

For the suitable wireless technology, there are many potential candidates. This selection depends on the nature of the application as well as constraints such as communication range, data rate etc., as shown in Table 2 [9,26–29]. For example, if the communication is between two IoT sensors near each other (less than 100 m range), Zigbee or Bluetooth can be used. For energy-sensitive applications, Zigbee or Bluetooth Low Energy (BLE) can be a good choice. For applications that require connectivity of around 100 m, IEEE 802.11 technology can be used to meet these requirements [30]. Similarly, if the required connectivity range of sensors is high (greater than 300 m), 6G communication is the most suitable option. Similarly, for applications such as vehicular networks, Cellular V2X or IEEE 802.11p standards are proposed by different working groups, industrial manufacturers,

and government organizations [31]. The 3rd Generation Partnership Project (3GPP) release 17 also contains the latest standards related to 5G communications [32].

**Table 2.** Selection of wireless technology based on application requirements.

| Communication Range (m) | Data Rate (Bits per s) | Wireless Technology |
| --- | --- | --- |
| 100 | 250 k | Zigbee |
| 60 | 1 M | BLE |
| 100 | 54 M | IEEE 802.11 |
| 100–1000 | 1 T | 6G |
| 100–500 | 10 G | Cellular V2X |

## 3. Major Use Cases of IoT

This section describes four major use cases of IoT and how data transmission works in these applications. As shown in Figure 1, the four important examples of IoT include the Internet of Vehicles (IoVs), Internet of Medical Things (IoMT), Industrial Internet of Things (IIoT), and Internet of Smart Grids (IoSG).

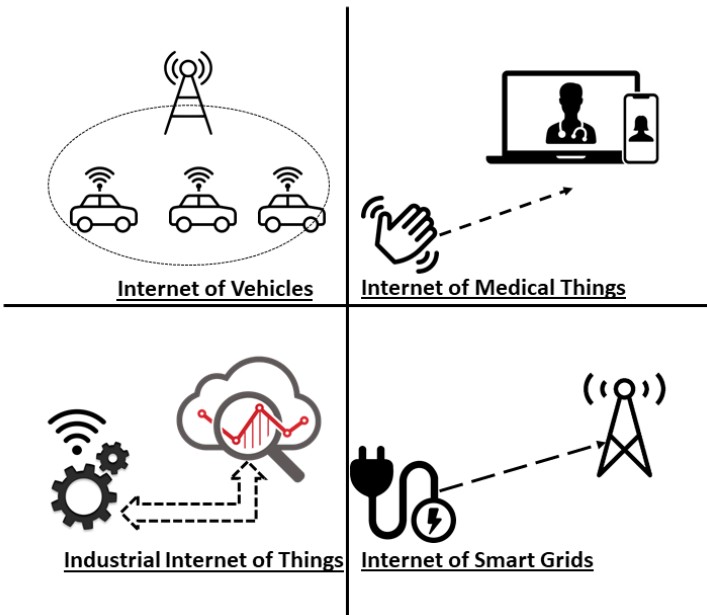

**Figure 1.** Challenges of Reliable Data Transmission in IoT.

### 3.1. Internet of Vehicles

Internet of Vehicles (IoVs) is a major application of IoT in the transportation industry [33–38]. IoVs involve vehicles that are equipped with wireless transceivers and infrastructure Road Side Units (RSUs) placed at different regions on the road. These transceivers enable communication between vehicles within a neighborhood, thus allowing vehicles to have a spatial map of the traffic. Vehicles can also share the mobility and traffic information with the RSUs, thus extending the range of communication [39–43].

In the IoV use case, traffic density is sensed by the number of vehicles within the communication range. RSUs can gather the traffic density information on the road through the vehicles or by evaluating the arrival rate of vehicles within a particular region. Other information that can be sensed includes safety awareness of vehicles, the emergency situation on the road, and traffic congestion.

IoVs can use IEEE 802.11p or Cellular V2X communications (C-V2X) to disseminate the sensed information to the cloud server located in RSUs or the central infrastructure unit of the city. While IEEE 802.11p uses ad hoc Wireless Local Area Network (WLAN)

based communications, C-V2X utilizes cellular communications to share and disseminate traffic data information [44,45].

In IoVs, the data analytic part involves finding out the useful insights that improve the traffic flow. Based on the traffic density and mobility information, RSUs can send safety-related notifications to the vehicles. In case of an emergency at a particular road, emergency departments such as hospitals and fire units can be informed. Similarly, efficient routes that facilities the drivers as well as reduce congestion on the roads can be evaluated [46,47].

### 3.2. Internet of Medical Things

Internet of Medical Things (IoMT) is a vital enabler for the future health industry. IoMT can provide applications such as remote patient monitoring, automatic monitoring of emergency patients, medical supply chain control, and contact tracing in pandemic situations [48–56].

In IoMT, the sensed data are related to patient body parameters such as heart rate, blood pressure, and sugar level. Several sensors can also help in the early detection of severe diseases such as cancers. Moreover, infectious diseases can also be detected earlier to control their spread [57–59].

The data dissemination in IoMT can be managed by long-range communication technologies such as 5G. The data, such as patient body parameters, must be shared periodically and reliably with the hospital monitoring room. In supply chain monitoring or contact tracing applications, security while disseminating the data also becomes a key concern.

The analyzed data in IoMT remote patient monitoring application is about patient health status. In case of a sudden change in a parameter, alarms can be generated, and doctors can be informed. Moreover, for supply chain monitoring, the temperature of the medicines or vaccines can be continuously checked [60–62].

### 3.3. Industrial Internet of Things

Industrial Internet of Things (IIoT) is an emerging use case of IoT. By connecting machines with the Internet, their health status can be regularly monitored, and repairs can be planned, thus reducing the outages. Moreover, many industrial processes can be automated and smartly controlled to increase productivity [63–70].

The sensed data in IIoT can be electrical current in a transistor, wear and tear of the machine and operating conditions of the industrial process. In addition, the productivity data of the industrial process and machine output can be sent to a cloud server [71–74].

The data dissemination in IIoT is generally handled using cellular communications. In scenarios where the power of the sensors is an issue, short-range and low-powered communication such as Zigbee can be used as first-hop [75,76].

Based on the received data, IIoT can provide predictive maintenance. In addition, analyzing the past values of the various parameters, changes in the operating conditions of the industrial processes can be made. In addition, the trend of the production values can indicate if any change is required to achieve the optimal yield of the industrial process.

### 3.4. Internet of Smart Grids

Internet of Smart Grids (IoSG) improves electricity usage and distribution by connecting the home devices with the grid. Smart meters can also be placed at homes that share electricity usage information with the grid. IoSG can reduce electricity usage, thus facilitating the customers as well as improving the carbon footprints and conserving electricity resources [77–82].

The primary data sensed in IoSG is the amount of electricity consumed in Kilowatt-hour (KWH). These data can be sensed at a home level or individual devices within a home level. For distribution, the electric loss is also an important parameter [83–87].

Cellular communications are mostly used to share electricity data from a smart meter to the electric grid. The data rate and latency requirements for this type of data may not be high. In this regard, 5G technology is the most suited to share electricity data periodically.

IoSG can analyze the electricity data and provide useful insights such as scheduling different loads at different times of the day. Moreover, it can predict the amount of bill based on current usage and allow users to take actions if needed [88–90].

### 3.5. Other Emerging Applications of IoT

Besides these four applications, there are many emerging use cases of IoT in areas such as process industry, agriculture, food and beverages industry, and supply chain. In the process industry such as petrochemical, continuous monitoring of key parameters of many equipment and machines is required. Examples of the important parameters include temperature, humidity, pressure and flow. By sensing these parameters and using data analytic techniques, predictive maintenance of machines can be realized. An example of such use case is measuring the temperature of cooling and heating fluid in a heat exchanger. Other machines such as centrifugal pumps, food storage containers, and winding machines can also use condition monitoring solutions to predict the faults and reduce the down time.

In agriculture, IoT can be used to efficiently utilize the available resources. Using the data collected by different sensors to monitor water levels, soil characteristics, rain fall, weather conditions, and crop yield, an intelligent machine learning model can be developed to predict crop health. Moreover, optimal use of disinfectant sprays and water can be figured out. In this regard, IoT based drones can also be used to collect data and also for automated water spraying.

### 4. Overview of Reliable Data Transmission in IoT Network

We provide an overview of reliable data transmission in IoT networks in this section. In Figure 2, we present three major components of reliable IoT data dissemination. These include resource allocation, latency management, and security.

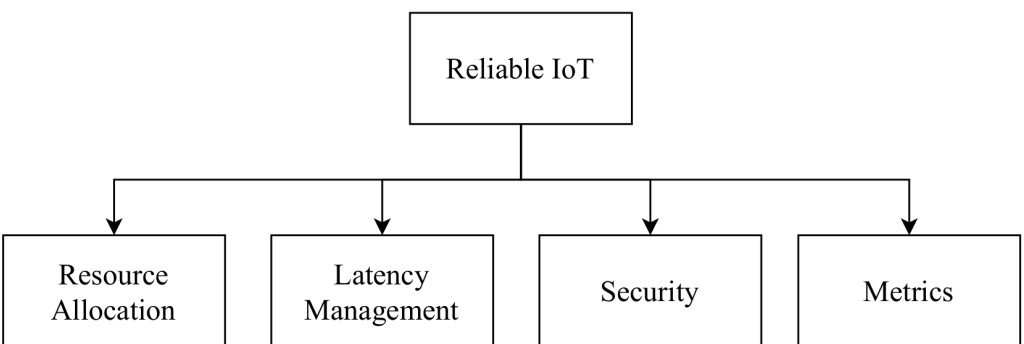

**Figure 2.** Key components of Reliable Data Transmission in IoT.

Efficient resource allocation is important for reliably sharing data between IoT nodes and servers. As the spectrum resource is limited due to large data generated by IoT nodes, it is important to propose intelligent spectrum utilization techniques. Techniques such as cognitive spectrum management could be used for sharing of spectrum bands by several IoT nodes. Transmit power is also an important resource that needs to be allocated carefully. To save energy of IoT nodes so that they can stay active for longer periods without charging, adaptive transmit power techniques are required.

Fog computing is a vital part of future IoT networks. Fog nodes located near the IoT devices provide storage and computational capacity to the IoT network. IoT networks can place popular and most useful content in the cache storage of these fog nodes. Hence, cache storage allocation is an important challenge. Moreover, IoT nodes may not perform all tasks locally and, therefore, will offload many tasks to these fog nodes. In this regard,

optimal task offloading algorithms are needed to ensure that the computational capacity resource of fog nodes is used efficiently.

Latency management is another crucial unit of reliable data transmission in IoT. IoT applications may not work well if data are not shared within the desired latency. Many new applications such as autonomous driving and industrial automation have stringent latency requirements, and hence, latency management is needed. In this regard, smart retransmissions can add diversity and enhance the probability of quick packet reception at the receiver. Moreover, optimal medium access techniques need to be developed to allow IoT nodes quick and fair channel access.

Accurate data traffic prediction can support latency management techniques as knowledge of upcoming traffic at an IoT server, and fog node equips it to handle it better. Hence, Artificial Intelligence (AI) based techniques that forecast the frequency and size of data can be very useful. Besides, other network technologies can support IoT networks to transmit their data quickly. In this regard, Unmanned Aerial Vehicle (UAV) and vehicular network has the potential to act as a relay for data traffic generated by IoT nodes [91–100].

Security is an essential component of reliable data transmission in IoT. Several attacks may be generated in an IoT network that can compromise the privacy and confidentiality of data transmitted. Moreover, malicious nodes can insert fake and wrong data in the network that can affect the decision-making of IoT applications. To tackle this issue, advanced cryptographic techniques are needed that can ensure the security of transmitted data while keeping the required overhead to a minimum. As an increase in the security overhead can result in higher data latency, tradeoff between security robustness and latency needs to be evaluated.

Blockchain is an upcoming technology that can provide robust security to IoT devices. Hence, blockchain-based solutions need to be developed in the context of IoT applications. Other security techniques such as physical layer security can also improve the reliability of IoT networks. These techniques can work in collaboration with cryptographic techniques to provide a robust solution. Lastly, data integrity attacks and anomaly detection schemes are also required to ensure that correct data are received based on which decisions can be made.

*Contributions of the Paper in Comparison with Other Survey Papers*

We present an overview of different recent survey papers related to IoT in Table 3. The survey paper in [101] focuses on security and privacy challenges in edge computing-based IoT. The authors categorize the different types of attacks in edge computing IoT and provide a list of solutions to mitigate these attacks. In [102], an overview of blockchain technology and its possible use cases in IoT is provided. Moreover, the authors highlight the blockchain applications for Industrial IoT (IIoT). The paper [103] provides an overview of different 5G enabling technologies such as Massive Input Massive Output (MIMO), mm-wave, and Heterogeneous networks (HetNets). The authors also review the security challenges for low-power 5G systems.

**Table 3.** Comparison of our survey paper in terms of contributions as compared to other IoT survey papers (RA = Resource Allocation, LM = Latency Management, SP = Security and Privacy, RM = Reliability Metrics, PL = Physical Layer Technologies and Techniques, EC = Edge Computing, ML = Machine Learning, EH = Energy Harvesting).

| Survey Reference | RA | LM | SC | RM | PL | EC | ML | EH | Survey Overview |
|---|---|---|---|---|---|---|---|---|---|
| [101] | ✗ | ✗ | ✓ | ✗ | ✗ | ✓ | ✗ | ✗ | Security & privacy challenges in Edge computing IoT<br>Categories of attacks in Edge computing IoT<br>Possible solutions to mitigate attacks |
| [102] | ✗ | ✗ | ✓ | ✗ | ✗ | ✗ | ✗ | ✗ | Blockchain opportunities in IoT<br>Blockchain applications for Industrial IoT |
| [103] | ✗ | ✗ | ✓ | ✗ | ✓ | ✗ | ✗ | ✗ | Overview of major 5G technologies<br>Multiple Input Multiple Output (MIMO)<br>mm-wave and Heterogeneous networks (HetNets)<br>Security issues in low power 5G systems |
| [104] | ✗ | ✗ | ✗ | ✗ | ✗ | ✓ | ✗ | ✗ | Caching techniques at the edge nodes<br>Edge nodes location selection techniques<br>Edge node content distribution techniques |
| [105] | ✗ | ✓ | ✗ | ✗ | ✓ | ✗ | ✗ | ✗ | Overview of major 6G technologies<br>Reconfigurable Intelligent Surfaces (RIS)<br>Terahertz communications<br>Major 6G applications |
| [106] | ✗ | ✗ | ✓ | ✗ | ✗ | ✗ | ✓ | ✗ | Identification of malicious and anomalous IoT devices<br>ML techniques for anomalous IoT devices |
| [107] | ✗ | ✗ | ✗ | ✗ | ✗ | ✓ | ✗ | ✗ | Survey of security techniques for IIoT applications<br>Fog computing opportunities to improve IIoT security |
| [108] | ✗ | ✗ | ✓ | ✗ | ✗ | ✓ | ✓ | ✗ | Significance of Federated Learning (FL) in IIoT<br>FL techniques for fog computing and security |
| [109] | ✗ | ✗ | ✓ | ✗ | ✗ | ✗ | ✗ | ✗ | Vulnerabilities data sources for IoT<br>Applicability & benefit of data for IoT evaluations |
| [110] | ✗ | ✗ | ✓ | ✗ | ✗ | ✗ | ✗ | ✓ | Categorization of attacks for IIoT<br>Security solutions for IIoT applications |
| [111] | ✗ | ✗ | ✗ | ✗ | ✗ | ✓ | ✗ | ✓ | Energy harvesting medium access techniques for IoT<br>MAC protocol for efficient energy harvesting |
| [112] | ✓ | ✗ | ✓ | ✗ | ✗ | ✓ | ✗ | ✗ | Overview of edge computing frameworks for IoT<br>Data management & security of edge computing IoT |
| Our Survey | ✓ | ✓ | ✓ | ✓ | ✗ | ✗ | ✗ | ✗ | Resource allocation techniques for IoT applications<br>Latency management techniques for IoT applications<br>Security techniques for IoT applications<br>Reliability metrics used for IoT evaluations |

The work in [104] reviews the caching techniques for edge computing IoT. In particular, the authors survey the edge node location selection techniques. In addition, edge node content distribution techniques are discussed. In [105], the authors present a discussion on upcoming 6G technologies such as Reconfigurable Intelligent Surfaces (RIS) and terahertz communications. Moreover, many 6G application scenarios are also discussed. The work in [106] explains the problem of identifying malicious and anomalous IoT nodes, which are a major security threat for IoT applications. The focus of the survey paper is on Machine Learning (ML) techniques to detect anomalous IoT devices and isolate them from the network.

A survey of recent security techniques useful for industrial settings is presented in [107]. The authors also discuss the use of fog computing to improve IIoT security. In [108], the authors present the significance of Federated Learning (FL) in IIoT. The authors discuss how distributed learning techniques can improve the security and computation cost of learning in IIoT. Various recent FL techniques for fog computing and security are surveyed. The work in [109] discusses the vulnerability data sources related to IoT. The paper surveys the applicability and benefit of data from these sources for IoT-related evaluations.

In [110], a categorization of various security attacks for IIoT applications is discussed. The authors also provide an overview of recent security solutions in this area. The work in [111] discusses energy harvesting-based medium access techniques for IoT. In particular, various Medium Access Control (MAC) protocols for energy harvesting systems are reviewed in detail. The paper in [112] presents an overview of different edge computing frameworks for IoT. The work also reviews different data management techniques and security techniques for edge computing IoT.

Compared to other survey papers, our work focuses on four key aspects of reliable data transmission in IoT. These include resource allocation techniques, latency management algorithms, security solutions, and reliability metrics for 6G based IoT. As shown in Table 3, IoT resource allocation has been reviewed in few papers such as [113]. Moreover, latency management techniques and reliability metrics have not been previously reviewed in survey papers, which is a uniqueness of our survey paper.

## 5. Review of Recent Work in Literature on Reliable Data Transmission in IoT

We present a brief review of recent work done in reliable data transmission in IoT networks. As shown in Figure 2, the four major areas related to reliability that are included in this paper are efficient resource allocation, latency management, security and reliability metrics. The following subsections discuss the literature review in these three key areas.

### 5.1. Resource Allocation

Resource allocation is an important area of research for IoT applications. Since IoT devices are constrained in energy, computation, and transmission, intelligent and novel resource allocation techniques are needed. As shown in Figure 3, resources such as spectrum, the transmission power of IoT nodes, cache storage of IoT enabled fog nodes, computational capacity of IoT nodes and fog nodes, and data rate needs to be carefully allocated. We present a review of recent work related to efficient resource allocation for IoT networks in Table 4. In [114], the authors consider a Wireless Powered Communication Network (WPCN) based wireless power transfer scenario. The communication between Access Point (AP) and mobile users is the focus of the paper. The work provides an optimal algorithm to allocated resources, including channel selection, transmission time, and transmit power. The joint optimization problem becomes non-convex and is solved using an iterative algorithm. Simulation results show improved sum-rate and reduced convergence time of the proposed algorithm.

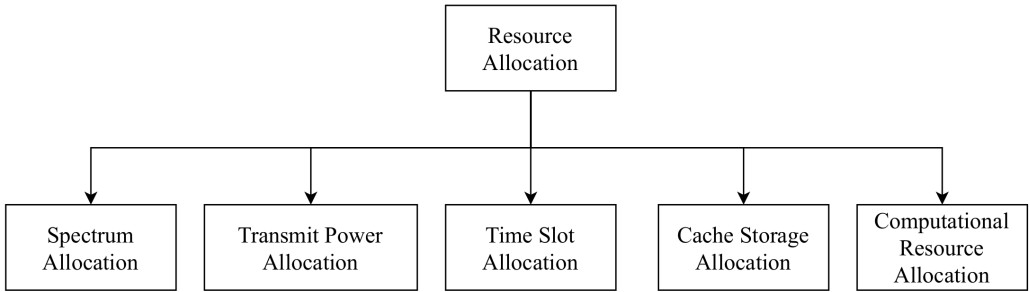

**Figure 3.** Different categories of recent work in Resource Allocation in IoT.

**Table 4.** Resource allocation in IoT: Recent literature review (S = Spectrum Allocation, P = Transmit Power Allocation, T = Time Slot Allocation, C = Cache Storage Allocation, M = Computational Resource Allocation).

| Scenario | S | P | T | C | M | Key Idea | Results |
|----------|---|---|---|---|---|----------|---------|
| Wireless power transfer [114] | ✗ | ✓ | ✓ | ✗ | ✗ | AP and mobile user communication<br>Optimal channel selection<br>Optimal time resource allocation<br>Optimal power resource allocation<br>Iterative algorithm for non-convex optimization | Improved sum-rate<br>Reduced convergence time |
| Edge IoT [115] | ✗ | ✗ | ✗ | ✗ | ✓ | IoT tasks offloaded to edge servers<br>Q-learning to allocate resources<br>Utility maximized and fairness<br>Online Q-learning scheme<br>Large state and action space<br>Reduce computation overhead<br>Reduce convergence time | Maximized application utility<br>Improved fairness |
| Smart factory [116] | ✓ | ✗ | ✗ | ✗ | ✗ | IoT enabled machines<br>Periodic data sharing with the server<br>Spectrum allocation in variable interference<br>Graph-Based Algorithm (GBA) used<br>Maximum weight matching in bipartite graphs | Increased transmitting users<br>Reduced transmission delay |
| Content-centric computing IoT [117] | ✗ | ✗ | ✗ | ✓ | ✗ | Improve Quality of Experience (QoE)<br>Factor such as Mean Opinion Score (MOS)<br>Cache resource allocation to improve QoE<br>Shortest Path Tree (SPT) algorithm<br>Deep-Q learning algorithm | Increased QoE<br>Reduced network cost |
| Energy self-reliant IoT network [118] | ✗ | ✓ | ✓ | ✗ | ✗ | Relay node harvests renewable energy<br>Relay node transmits data<br>Relays provide RF power to IoT nodes<br>Solved time and power resource allocation<br>Lyapunov optimization for max throughput | Increased data rate<br>Increased throughput |
| UAV network supporting IoT [119] | ✓ | ✓ | ✗ | ✗ | ✗ | IoT devices grouped into equal-sized clusters<br>Matching theory-based algorithm<br>Match UAV sub-channels to IoT nodes<br>Interference is minimized<br>Alternate optimization used<br>Optimized placement of UAV nodes<br>Optimized transmit power of IoT nodes | Reliable power selection<br>Reduced power of IoT nodes |

In [115], the authors consider an edge IoT scenario where IoT devices offload their tasks to nearby edge nodes. The authors use machine learning algorithms for efficient task offloading. A Q-learning-based algorithm is developed to allocate computational resources. The goal of the protocol is to maximize the utility of applications and achieve fairness. Moreover, an online Q-learning scheme to handle large state and action space is proposed, which reduces computation overhead and convergence time. The performance evaluation shows maximized IoT application utility and improved resource allocation fairness.

The authors in [116] consider a smart factory scenario where IoT-enabled machines periodically share data with the server. These data can include different parameters of the machines. The work proposes a spectrum allocation technique in the case when bands are impacted by interference differently. A Graph-Based Allocation (GBA) algorithm is used for resource allocation. The spectrum allocation problem is formulated as a bipartite graph

ad successive maximum weight matching is usedThe results show an increased number of users that transmit data. In addition, data transmission delay is considerably reduced.

The work in [117] considers a content-centric IoT scenario where devices offload their data to cache storage of fog nodes. The goal of the two proposed techniques is to improve user satisfaction in terms of Quality of Experience (QoE) by using efficient cache resource allocation. Factors such as the Mean Opinion Score (MOS) are considered while computing the QoE. The first technique uses Shortest Path Tree (SPT) algorithm, whereas the second technique uses the Deep Q-learning algorithmThe results show an increase in QoE and reduced network cost.

In [118], an energy self-reliant IoT network is considered. Relay nodes are chosen in the network that harvests energy using renewable sources. The relay node serves two purposes, enabling data transmission by forwarding signals from the source to the destination and powering multiple IoT nodes using RF signals. The authors solve the time and resource allocation problem using Lyapunov optimizationThe results show an increased achievable data rate and throughput.

The work in [119] considers a UAV network that supports IoT devices in terms of providing computation resources. IoT devices are grouped into equal-sized clusters. A matching theory-based algorithm is proposed that matches UAV sub-channels to IoT nodes such that interference is minimized. The work uses alternate optimization for optimal placement of UAVs and transmit power selection of IoT nodesThe results show improved reliability to achieve optimal transmit power. Moreover, IoT nodes use lower transmit power values when using the proposed algorithm.

### 5.2. Latency Management

Latency is a vital QoS indicator for IoT applications. Most applications are sensitive to latency and need a lower latency value within a threshold to ensure reliable communication. As shown in Figure 4, techniques such as smart retransmissions, resource allocation, multiple access and physical layer techniques, traffic prediction, and cooperation from other networks is used for improving the latency of IoT. This subsection provides an overview of work done in latency management in IoT as shown in Table 5.

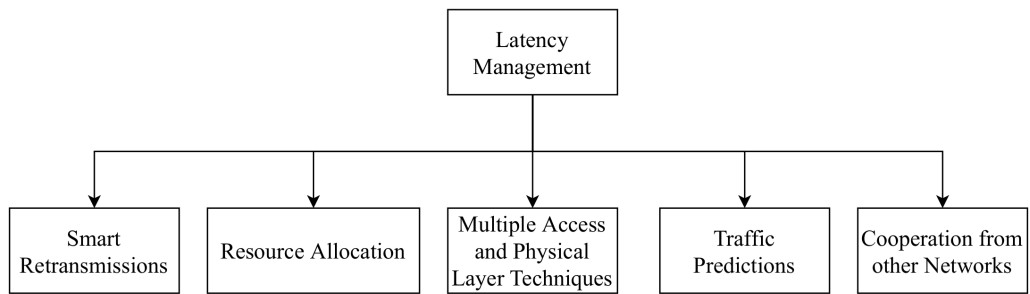

**Figure 4.** Different categories of recent work in Latency management in IoT.

The authors in [120] considered a narrowband IoT (NB-IoT) scenario where an increased number of retransmissions in the standard improves the packet delivery ratio but increases the energy consumption and latency. The proposed technique computes optimal durations of the control channel and data channel. Moreover, the authors also find out the optimal coverage class for each user where coverage class defines the number of retransmissions allowed. Simulation results highlight reduced energy consumption and reduced latency achieved by the proposed protocol.

**Table 5.** Latency management in IoT: Recent literature review (R = Smart Retransmissions, A = Resource Allocation, M = Multiple Access and Physical Layer Techniques, P = Traffic Prediction, O = Cooperation from other networks such as UAVs).

| Scenario | R | A | M | P | O | Key Idea | Results |
|---|---|---|---|---|---|---|---|
| NB-IoT [120] | ✓ | ✗ | ✓ | ✗ | ✗ | Retransmissions impact on energy and latency<br>Optimal control & data channel duration selection<br>Optimal selection of coverage class per user | Reduced energy consumption<br>Reduced latency |
| multiple RATs to connect with MEC [121] | ✗ | ✓ | ✗ | ✗ | ✗ | Optimal task offloading decisions<br>Amount of task locally computed<br>Amount of task computed at MEC server<br>Task splitting between MEC servers<br>Optimal transmit power selection<br>Optimal RAT selection for task offloading<br>Alternate Convex Search (ACS) algorithm | Reduced energy consumption<br>Reduced latency |
| Cloud with several VNFs [122] | ✗ | ✓ | ✗ | ✓ | ✗ | VNFs are mapped to different applications<br>VNFs allocation to reduce latency<br>Regression for predicting application demand<br>Optimal allocation of VNFs as per demand<br>Initial offline training followed by online training | Reduced latency<br>Improved prediction accuracy |
| Terrestrial networks, satellite IoT and MEC [123] | ✗ | ✓ | ✗ | ✗ | ✗ | Data from IoT to satellites and gateways<br>Satellites have energy constraints<br>Data offloaded to gateways for processing<br>Optimal IoT association with a satellite & a gateway<br>Lagrange multiplier and DRL used | Reduced latency<br>Reduced energy cost |
| UAV assisted MEC based IoT [124] | ✗ | ✓ | ✓ | ✗ | ✓ | UAV facilitates storage and computation<br>UAV improves network coverage using relaying<br>Problem divided into three subproblems<br>First, IoT node and UAV association<br>Second, communication resource scheduling<br>Third, the UAV placement | Reduced latency |
| Semi-blind downlink NOMA based IoT [125] | ✓ | ✗ | ✗ | ✗ | ✗ | Improve SIC technique<br>Interference AICA used | Improved signal to noise ratio<br>Reduced latency |

The work in [121] considers a scenario where IoT devices use multiple Radio Access Technologies (RATs) to transmit data to the Mobile Edge Computing (MEC) server. The proposed protocol computes optimal task offloading decisions, including the number of tasks locally computed at IoT device, amount of task computed at MEC server, and task splitting between different MEC servers. Besides, optimal transmit power and RAT selection for task offloading are also proposed. The developed optimization problem is solved using the Alternate Convex Search (ACS) algorithm. The results highlight reduced energy consumption as well as reduced latency when the proposed protocol is used.

In [122], the authors consider a cloud scenario with several Virtual Network Functions (VNFs). VNFs are mapped to different applications such as delay-sensitive, mission-critical, and latency tolerant. The goal of the proposed technique is the efficient allocation of VNFs to reduce latency. The authors use a regression algorithm to predict the demand of each application type and make allocations accordingly. The proposed algorithm works initially on offline training. This is followed by online training afterward.

The work in [123] considers a terrestrial network scenario composed of IoT-enabled satellites and a MEC server. IoT nodes collect data and send it to satellites or satellite gateways for further processing. As satellites have energy constraints, they offload the IoT data to the gateways for further processing. The proposed work finds an optimal association between a satellite and a gateway using Lagrange multiplier and Deep Reinforcement Learning (DRL) techniquesThe results highlight reduced latency of IoT data at a reduced energy cost.

In [124], a UAV-assisted MEC-based IoT network scenario is considered. UAV facilitates computing in terms of storage and computation service. UAV can also improve network coverage by providing relaying services to overcome poor channel conditions. The problem is divided into three sub-problems. The first one deals with IoT nodes and UAV association, the second one handles scheduling of communication resources, and the last one manages UAV placement. The proposed algorithm shows low latency as compared to other related techniques.

The work in [125] considers a semi-blind Non-Orthogonal Multiple Access (NOMA) based IoT scenario. The proposed technique uses an interference Alignment Independent Component Analysis (AICA) as compared to the traditional Successive Interference Cancellation (SIC) techniqueThe results highlight improved signal-to-noise ratio and reduced latency.

*5.3. Security*

Security is an important requirement for reliable data transmission in IoT. Techniques that provide defense against malicious users and their attacks are required to ensure reliability in IoT. As shown in Figure 5, security techniques include cryptography techniques, blockchain-based techniques and data integrity detection techniques. In Table 6, we provide the recent work related to security in IoT.

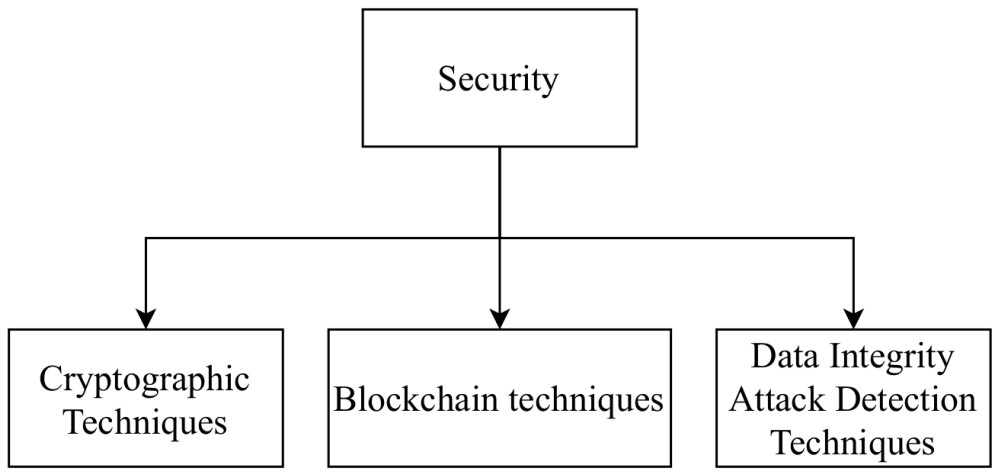

**Figure 5.** Different categories of recent work in Security in IoT.

**Table 6.** Security in IoT: Recent literature review (C = Cryptographic Technique, B = Blockchain, D = Data Integrity Attack Detection).

| Scenario | C | B | D | Key Idea | Results |
|---|---|---|---|---|---|
| Anomaly detection at edge gateways [126] | ✗ | ✗ | ✓ | Classify traffic as anomalous and normal based Fuzzy C-means clustering & fuzzy interpolation Access to malicious IoT nodes restricted | Improved accuracy Reduced false positive rate |
| Transport layer security for IoT applications [127] | ✓ | ✗ | ✗ | Reduce latency and achieve forward secrecy Identity-based cryptographic technique Identity-based encryption for client-server data | Reduced latency Reduced traffic overhead |
| Secure 5G Internet of drones [128] | ✗ | ✓ | ✗ | Blockchain for secure transmission Private blocks & Transactions are recorded Novel consensus algorithm is developed | Robustness against attacks Lower communication overhead Lower computation time |
| IoT insider attacks [129] | ✗ | ✗ | ✓ | Malicious attack detection from IoT insiders AI and distance based attacks classification | Improved accuracy Reduced computation time |
| AES for constrained IoT devices [130] | ✓ | ✗ | ✗ | Reduce complexity of the standard AES algo. Use a reduced number of rounds for algorithm Mathematical proof of proposed protocol | Reduced encryption time |
| Data Integrity attacks in IoV [131] | ✗ | ✗ | ✓ | Isolation forest algorithm used Find anomalies in traffic density information Detected anomalies verified Verification from the neighborhood area | Improved anomaly detection accuracy Reduced false positives |

The work in [126] proposes an anomaly detection technique for IoT devices. The proposed security algorithm is implemented at edge gateways and classifies traffic as abnormal and normal. The classification is done using fuzzy C-means clustering and fuzzy interpolation. Once the anomaly is detected, access to the malicious IoT nodes is

restrictedThe results signify the performance of the proposed protocol in terms of improved accuracy of anomaly detection and reduced false-positive rate.

In [127], the authors proposed a transport layer security protocol for IoT applications. The goal is to reduce the latency and achieve forward secrecy. In this regard, an identity-based cryptographic technique is proposed. The key idea is that the client uses identity-based encryption to transmit data to the server when it receives no response from the serverThe results show reduced latency and reduced traffic overhead.

The work in [128] proposes a secure communication technique for 5G Internet of drones. Blockchain technology is used to ensure secure data transmission. Private blocks are created, and transactions are recorded in them. Moreover, a novel consensus algorithm is also developed. The result shows that the proposed protocol provides robustness against attacks, lower communication overhead, and lower computation time.

In [129], the authors propose a security protocol to detect malicious attacks which are from inside the IoT network. Artificial Intelligence (AI) based distance measurement techniques are used to identify and classify malicious attacksThe results show improved accuracy of detection and reduced computation time.

The authors in [130] provide Advanced Encryption Standard (AES) based security algorithm for constrained IoT devices. The proposed protocol reduces the complexity of the standard AES algorithm by using a reduced number of rounds. The mathematical proof of the proposed protocol is also providedThe results indicate reduced encryption time of the proposed protocol.

In [131], the authors present a technique to detect data integrity attacks in the Internet of Vehicles (IoVs). In this regard, the isolation forest algorithm finds anomalies in the traffic density information shared by vehicles. After detecting anomalies, verification messages are sent to neighborhood vehicles of potential malicious vehicles to verify if the data transmitted is malicious or correct. Simulation results show that improved accuracy of anomaly detection and reduced false positives.

### 5.4. Reliability Metrics for IoT

In this subsection, we discuss different reliability-related metrics used to evaluate IoT applications' performance. We present a list of these metrics in Figure 6, and provide their key idea and uses in Table 7.

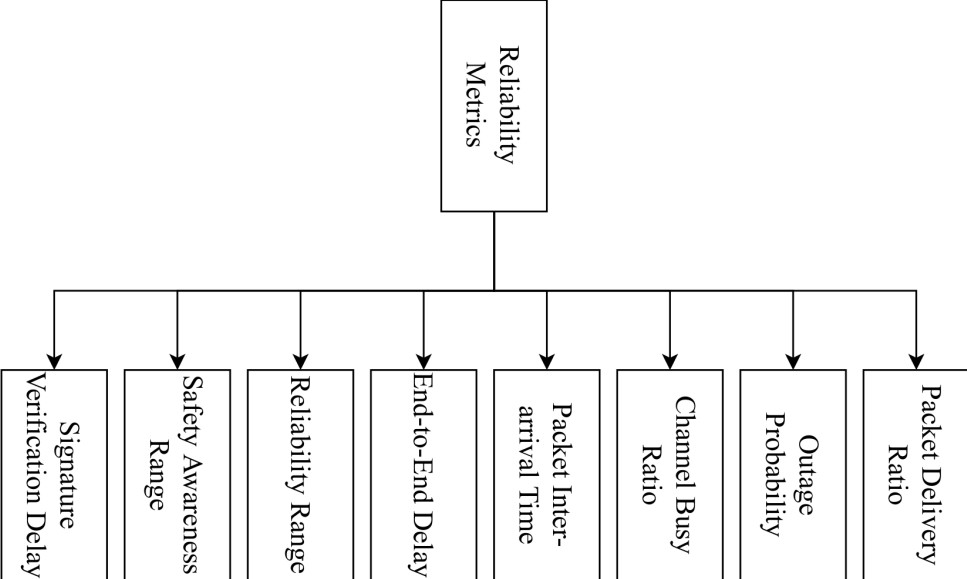

**Figure 6.** Different Reliability Metrics used in IoT.

Packet Delivery Ratio (PDR) is a commonly used reliability metric that measures the ratio of the total number of packets received at a receiver and the total number of packets transmitted by the transmitter [132]. Most IoT applications require a very high PDR value as it ensures that the data between the nodes is shared without any error. With 6G communications, many novel technologies and techniques are used that enhance the PDR to a value of 99.99999%. A low PDR value implies that the channel conditions between the transmitter and receiver are not good due to factors such as multi-path fading. Thus PDR value can provide important feedback on allocating resources such as transmission power of the nodes. A high transmit power may be used for scenarios that suffer from low PDR.

Outage probability is another useful metric for resource allocation as it indicates the distance at which the transmitter and receiver become out of range [133–136]. As a result, the node can select the optimal transmit power and modulation scheme based on IoT application requirements. A higher transmit power will increase the transmission distance, thus mitigating the channel impairments. On the other hand, a lower modulation scheme reduces the data rate but increases the transmission distance.

**Table 7.** Reliability metrics used for IoT.

| Reliability Metric | Metric Key Idea | Uses |
|---|---|---|
| Packet Delivery Ratio | Ratio of packets successfully received | Resource allocation IoT |
| Outage Probability | Receiver becomes out of range of transmitter | Resource allocation IoT |
| Channel Busy Ratio | Measure of channel load in the network | Resource allocation IoT |
| Packet Inter-arrival time | Time difference between two consecutive packets at receiver | Latency management IoT Vehicular Networks |
| End-to-End Delay | Total time taken to receive a packet | Latency management IoT |
| Reliability Range | Communication range at which packets are received with a probability gr | Vehicular Networks |
| Safety Awareness Range | Vehicle safety and message reliability for finding reliable range | Vehicular Networks |
| Signature verification time | Total time required to verify the signature of a message | Security IoT |

Another important reliability metric for IoT is the channel busy ratio, which indicates the total data load on the network [137]. This metric can be measured by the IoT radio based on the percentage of time the radio senses the channel as free. Resource allocation can be optimized based on channel busy ratio measurements, such as data generation rate, transmit power, and modulation scheme.

Packet inter-arrival time is another metric that computes the time difference between two consecutive packets at the receiver [138]. Most IoT applications require a certain packet inter-arrival time threshold, so that information shared by IoT sensors does not expire. A variable packet inter-arrival time can result in unwanted delays in sensor measurement dissemination to the server, reducing the application data analysis accuracy.

The end-to-end delay of the packet is a crucial metric that provides information about how much delay is required to transmit the packet [138]. As discussed before, a higher end-to-end delay can be caused by poor channel conditions, network congestion or inefficient multiple access. Latency management techniques can be evaluated based on the end-to-end delay metric.

Another useful metric for the Internet of Vehicles (IoVs) is the safety awareness range [139]. This metric uses information accuracy of the messages shared between vehicles to assess how far vehicles can connect reliably. Information shared between vehicles includes their position, acceleration, speed, etc. As other reliability metrics do not focus on information accuracy, this metric provides a better measure of reliability, especially for vehicular networks.

Signature verification time is another important metric that measures how much time it takes to process the message in terms of security at the receiver [140]. As many packets

are received at a receiver within a short time, they may have to wait in a queue before going through the signature verification process. A higher signature verification time means that overall end-to-end delay is increased. Hence, nodes may have to use another security algorithm with low encryption key sizes, or a faster processor is needed at the receiver.

## 6. Future Opportunities and Challenges

We present future opportunities and challenges to enable reliable IoT applications in this section. We discuss three important opportunities that can enhance the reliability of future IoT applications. These opportunities include machine learning techniques, 6G communications, and blockchain-enabled security.

### 6.1. Machine Learning Techniques

As future IoT applications will generate a large amount of data, intelligent machine learning techniques are needed to analyze the data and get useful insights to improve the reliability of IoT [141]. Many machine learning techniques can be useful for IoT applications. We provide an overview of few machine learning techniques and their use cases in IoT in Figure 7.

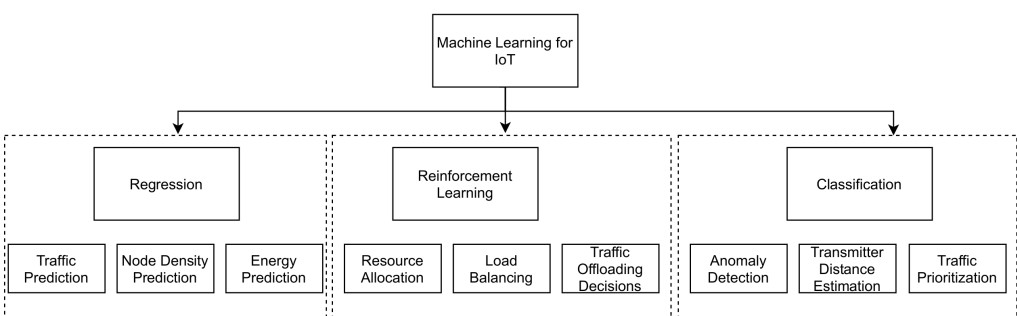

**Figure 7.** Future opportunities of using Machine learning for IoT.

Regression-based techniques are useful for predicting many important parameters in IoT [142]. One application of regression is to predict the data traffic and load on the network. By accurately predicting the traffic load, optimal resource allocation can be realized. Moreover, traffic prediction can be used to develop efficient congestion control techniques. Another application of traffic prediction is optimal load balancing for fog computing IoT networks. Using the predicted load (in terms of the number of tasks received for computation) on different fog nodes, tasks can be fairly distributed to the fog nodes.

Vehicle density prediction in an IoV can also utilize regression techniques. Infrastructure Road Side Units can collect large data of vehicle mobility and predict future vehicle densities at different times of the day. Another key application of regression techniques is the energy prediction of different IoT nodes. The energy of IoT nodes depends on factors such as node's idle time duration, transmission time duration, and transmission power. A robust prediction of these parameters allows IoT applications to take corrective actions. An accurate energy prediction technique help design robust energy-aware algorithms that can enhance the lifetime of IoT nodes and also reduce chances of total energy depletion.

Reinforcement learning techniques can also be used in IoT for better resource allocation and load balancing [143]. Using reinforcement learning algorithms, optimal actions such as transmission power selection, cache placement in the fog nodes, and task offloading ratios can be figured out. The reward function in these algorithms can be based on wireless interference, network sum-rate, task offloading time and cache access time.

Classification techniques such as k-nearest neighbor and decision trees, etc., can be used to solve challenges such as anomaly detection for enhanced security. Malicious nodes can launch attacks on IoT nodes in the form of false data sharing or jamming signals.

Thus, it is vital to detect abnormal traffic to maintain the reliability of the IoT network. Another use of classification techniques is for transmitter-receiver distance estimation. Techniques such as k-means clustering can be used in this regard to find how far the receiver is located [144]. In addition, data traffic can be classified into various classes based on their QoS requirements to design priority algorithms.

### 6.2. 6G Communications

6G communications will be an important enabling technology for future IoT applications. By improving the communications in terms of achievable data rate, packet delivery ratio, and latency, reliable data dissemination between IoT nodes can be achieved. 6G will use terahertz communications, Reconfigurable Intelligent Surfaces (RIS), and massive computing to significantly improve the end-to-end communications between various IoT-enabled nodes.

Terahertz communications can provide short-range high data rate connectivity between IoT nodes. Massive IoT applications will require increased data transmissions, and terahertz communications can be an ideal choice. However, challenges such as the connectivity range of terahertz for IoT need further investigation [145]. To further enhance the range of terahertz, techniques such as multi-hop communication can be used. Another potential technology to improve the range is reconfigurable intelligent surfaces that can intelligently reflect signals towards the destination [146]. Challenges such as RIS selection for an IoT transmitter-receiver pair and RIS phase shift optimization require further attention [147].

Massive computing algorithms are also needed as part of 6G communications. In this regard, challenges such as placement of fog nodes, computational resource allocation, and task offloading ratios need to be designed. Moreover, other factors such as the energy of the fog nodes is a critical factor that needs to be considered.

### 6.3. Blockchain-Enabled Security

Robust security mechanisms need to be developed for IoT applications. Some IoT applications are critical such as health monitoring, vehicle safety data sharing, etc. Hence, attacks on these applications can cause safety concerns to humans. Blockchain is an efficient technique that can ensure secure data communications due to its distributed record-keeping and proof-of-work mechanism [148,149].

While blockchain technology has many use cases in IoT, major challenges such as mining task computation and consensus algorithm selection need further research. Moreover, the energy required to mine the blockchain transactions is a critical challenge for which energy-efficient algorithms need to be designed. IoT can also benefit from blockchain-based smart contracts, which have applications in the supply chain, healthcare, and industrial automation. However, further studies are required to analyze the performance and security of a blockchain-enabled smart contract [150,151].

## 7. Conclusions

In this paper, we survey the current work and future opportunities related to reliable data transmission in IoT. We present four key components of reliable data sharing in the context of IoT, which include resource allocation, latency management, security and reliability metrics. We discuss the techniques and algorithms that have been proposed to ensure reliability in IoT. We also highlight the major challenges that still need attention for implementing reliable future IoT networks.

**Author Contributions:** This article was prepared through the collective efforts of all the authors. Conceptualization, M.Z.K., O.H.A., M.A.J., H.G. and K.S.A.; Critical review, M.Z.K., O.H.A., M.A.J., H.G. and K.S.A.; Writing—original draft, M.Z.K., O.H.A. and M.A.J.; Writing—review & editing, H.G. and K.S.A.All authors have read and agreed to the published version of the manuscript.

**Funding:** The authors extend their appreciation to the Deputyship for Research & Innovation, Ministry of Education in Saudi Arabia for funding this research work the project number (442/84). Also, the authors would like to extend their appreciation to Taibah University for its Supervision support.

**Institutional Review Board Statement:** Not applicable.

**Informed Consent Statement:** Not applicable.

**Conflicts of Interest:** The authors declare no conflict of interest.

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
