# Peer review of "Reliable Internet of Things: Challenges and Future Trends"

_electronics, doi:10.3390/electronics10192377_

Round 1

Reviewer 1 Report

Please find my comments below:

  1. Since this is a review paper, it is recommended to add details about some emerging use cases of IoT such as IoT in agriculture. I understand that the authors already mentioned the 4 major fields. However, there are many other review articles on the same. It would be more interesting to see some emerging applications.
  2. It is advised to elaborate on subsections 2.2- 2.5.  The heading of this section is Data Transmission in IoT. However, the subsections do not provide adequate details. 
  3. The headings of the sections are kind of repetitive and not informative enough. For instance, Sections 2, 3, and 4 all are somehow using the same title 'Data Transmission in IoT'.  The information provided in different sections is also repetitive. 

Author Response

Response to reviewer comments attached as pdf. 

Reviewer 2 Report

The authors propose a survey on four important IoT thematic cores (resource allocation, latency management, security and reliability metrics); Furthermore, they highlight important challenges (still under investigation) to have a reliable IoT system: application of machine learning techniques, 6G communications and blockchain-based security.

Major Comments:
1. The authors in section 2.1 have made an overview of the useful requirements for having a mass application of IoT systems, it would be useful to insert bibliographic references that confirm the data in tables 1 and 2.
2. The authors should include modern bibliographic references that illustrate the basic concepts of machine learning for the IoT and in particular the techniques based on regression, reinforcement learning and classification.
3. Authors should include modern references to support what is described in section 5.2, where they illustrate the basic technologies for realizing 6G communication.
4. Authors should include modern references to support what is described in section 5.3, where they illustrate the security mechanisms for blockchain-based IoT applications.

Minor Comments:

1. Authors should pay attention to the journal template, for example, in the text the figure must be cited as “Figure” and not “Fig.”.
2. Figure 1 for greater fluency in reading and understanding the text should be moved to section “2. Data transmission in IoT ". While, Table 1 and Table 2 should be inserted in section 2.1.
3. Information is missing (volume, date and number of pages) for references 27, 56-61, 63, 68, 97-98, 100.
4. Authors should arrange the text found in table 4, sixth row and seventh column (some text fills the adjacent column).
5. Figure 4 and Table 5 should be moved to section 4.2 for greater understanding and reading of the text.
6. Authors should insert figure 5 before table 6 (as it is cited first in the text).
7. Authors should better optimize the space between the text and the table and / or figure on page 13 and also on pages 15 and 16.

Author Response

(The authors gave the same response as above.)

Reviewer 3 Report

The paper is devoted to an overview of future IoT applications with a focus on communication requirements, resource allocation, latency management, security, and reliability metrics. The following IoT-based applications are considered and used as examples and cases in the paper: Internet of Vehicles, Internet of Medical Things, Industrial Internet of Things, and Internet of Smart Grids. Analysis of other state-of-the-art review papers in the same research field is provided. A review of relevant recent works on IoT is given, more than 100 references were analyzed during the preparation of this paper.
There are some issues in the paper that should be addressed by the authors so as to improve paper quality:
(1) Table 1 contains a requirement of 99.99999% to the reliability, but doesn’t describe which reliability metric or indicator is considered for this requirement;
(2) Statements given in Table 2 do not have reasoning or support information: what do “Low”, “Medium” or “High” mean? Which approach or technique was used to prepare this table?
(3) Section 4.4 lacks calculation expressions for provided metrics and references to normative documents (if any) or publications that establish and describe these metrics; some real, typical, or required values of these indicators for Internet of Vehicles, Internet of Medical Things, Industrial Internet of Things, and Internet of Smart Grids would also significantly simplify understanding;
(4) Paper lacks an overview of relevant normative documents in the IoT field, communications, reliability, security, etc. – review only of publications was performed;
(5) 5G applications are mentioned throughout the paper. It would be helpful if analysis of transmission to 6G was added, covering new challenges, risks and opportunities;
(6) Paper mentions IEEE802.11, but the document itself (https://ieeexplore.ieee.org/document/9363693) and IEEE 802.11p amendment are not given in references;
(7) extra space before Table 7 on Page 16 should be eliminated;
(8) typo: extra “1” should be removed before “1Department of Computer Science, College of Computer Science and Engineering, Taibah University”
(9) typo: Table 6 title doesn’t have “)”
(10) some used abbreviations should be disclosed: UAV etc.

Author Response

(The authors gave the same response as above.)

Round 2

Reviewer 1 Report

Thanks for addressing my earlier comments

Reviewer 3 Report

The authors provided significant changes in the revised version taking into account all comments, thus I'll recommend the paper to be published.